# Snail reprograms glucose metabolism by repressing phosphofructokinase PFKP allowing cancer cell survival under metabolic stress

Nam Hee Kim[1,*], Yong Hoon Cha[1,*,†], Jueun Lee[2,3], Seon-Hyeong Lee[4], Ji Hye Yang[1], Jun Seop Yun[1], Eunae Sandra Cho[1], Xianglan Zhang[1,5], Miso Nam[2,3], Nami Kim[2], Young-Su Yuk[1], So Young Cha[1], Yoonmi Lee[1], Joo Kyung Ryu[1], Sunghyouk Park[6], Jae-Ho Cheong[7], Sang Won Kang[8], Soo-Youl Kim[4], Geum-Sook Hwang[2], Jong In Yook[1] & Hyun Sil Kim[1]

Dynamic regulation of glucose flux between aerobic glycolysis and the pentose phosphate pathway (PPP) during epithelial–mesenchymal transition (EMT) is not well-understood. Here we show that Snail (SNAI1), a key transcriptional repressor of EMT, regulates glucose flux toward PPP, allowing cancer cell survival under metabolic stress. Mechanistically, Snail regulates glycolytic activity via repression of phosphofructokinase, platelet (PFKP), a major isoform of cancer-specific phosphofructokinase-1 (PFK-1), an enzyme involving the first rate-limiting step of glycolysis. The suppression of PFKP switches the glucose flux towards PPP, generating NADPH with increased metabolites of oxidative PPP. Functionally, dynamic regulation of PFKP significantly potentiates cancer cell survival under metabolic stress and increases metastatic capacities in vivo. Further, knockdown of PFKP rescues metabolic reprogramming and cell death induced by loss of Snail. Thus, the Snail-PFKP axis plays an important role in cancer cell survival via regulation of glucose flux between glycolysis and PPP.

[1] Department of Oral Pathology, Oral Cancer Research Institute, Yonsei University College of Dentistry, Seoul 03722, Korea. [2] Integrated Metabolomics Research Group, Western Seoul Center, Korea Basic Science Institute, Seoul 03760, Korea. [3] Department of Chemistry, Sungkyunkwan University, Suwon 16419, Korea. [4] Cancer Cell and Molecular Biology Branch, National Cancer Center, Ilsan 10408, Korea. [5] Department of Pathology, Yanbian University Medical College, Yanji City, Jilin Province 133000, China. [6] College of Pharmacy, Natural Product Research Institute, Seoul National University, Seoul 08826, Korea. [7] Department of Surgery, Yonsei University College of Medicine, Seoul 03722, Korea. [8] Department of Life Sciences, Research Center for Cell Homeostasis, Ewha Womans University, Seoul 03760, Korea. * These authors contributed equally to this work. † Present address: Department of Oral and Maxillofacial Surgery, Yonsei University College of Dentistry, Seoul 120-752, Korea. Correspondence and requests for materials should be addressed to G-S.H. (email: gshwang@kbsi.re.kr) or to J.I.Y. (email: jiyook@yuhs.ac) or to H.S.K. (email: khs@yuhs.ac).

Cancer cell survival under metabolic stress is a critical step not only for solid tumour growth but also metastatic progression[1–3]. While the glucose in cancer cells is mainly metabolized through aerobic glycolysis to provide biomass, glucose flux into the pentose phosphate pathway (PPP) is an important metabolic circuit generating NADPH (nicotinamide adenine dinucleotide phosphate, reduced)[3–5]. In a resource-limited environment such as matrix detachment or lack of glucose supply, cancer cells suspend their anabolic glycolysis and reprogram metabolic homeostasis, surviving mainly on mitochondrial oxidative phosphorylation[3,6]. Therefore, maintaining an adequate level of NADPH is critical for cancer cells to survive oxidative stress[7,8]. Indeed, many oncogenic signalling networks play an important role in regulating PPP flux and NADPH homeostasis. For example, p53-induced TIGAR (identified as fructose-2,6-bisphosphatase) reduces glycolytic activity and promotes PPP[9,10]. Conversely, p53 itself inhibits G6PD, a gatekeeper enzyme of PPP, and loss of p53 function increases glucose flux towards PPP[11]. LKB-AMPK activation prolongs cancer survival under metabolic stress via suppression of NADPH consumption[4]. However, the oncogenic role of NADPH homeostasis via PPP during cancer progression has not been widely studied.

The transcriptional repressor Snail triggers epithelial–mesenchymal transition (EMT), allowing cancer cells with invasive and stemness properties[1,12]. Major oncogenic signalling pathways such as canonical Wnt and p53 tumour suppressor modulate Snail activities[13–15], indicating the key role of Snail abundance during cancer progression. Although earlier studies reinforced phenotypic conversion and migratory potential during EMT, recent evidence suggests that EMT of cancer cells is involved in therapeutic resistance and cancer stemness[16,17]. While overcoming metabolic stress is critical for cancer progression as well as for therapeutic resistance, the underlying mechanism of Snail in metabolic reprogramming is not well-understood.

The conversion of fructose-6-phosphate (F6P) to fructose-1,6-bisphosphate (F1,6BP) catalysed by PFK-1 is the most important control step in the mammalian glycolytic pathway, while the oncogenic regulation of PFK-1 and its consequences are not well known. In this study, we found that Snail functions as a metabolic switch between aerobic glycolysis and PPP by repressing PFKP, a cancer-specific PFK-1, allowing cancer cell survival under metabolic stress. This study therefore provides molecular mechanisms whereby metabolic reprogramming overcomes oxidative stress during cancer progression.

## Results

**Snail supports cell survival under metabolic stress**. During the cancer progression, the cancer cell should not only proliferate in a nourished environment, but also survive in a resource-limited condition. Given its key function during cancer invasion and metastasis, we investigated whether Snail abundance is required for breast cancer cell survival under metabolic stress and proliferation in nourished condition. In normal culture condition, the lack of Snail slightly increased cancer cell proliferation (Supplementary Fig. 1a) while its overexpression led to enriched G0/G1 fraction with slowing cell growth[17] (Supplementary Fig. 1b,c). To test whether Snail abundance is required for cancer cell survival under resource-limited conditions, we compared cell death in a glucose-starved condition and the clonogenic capacity of the survived cells according to the Snail status. Contrary to in a nourished environment, knockdown of Snail rendered cancer cells more sensitive to cell death induced by glucose deprivation (Fig. 1a). To examine their clonogenic capacity, starved cells were refreshed with normal culture

medium for 2 weeks until colony formation was observed. Indeed, Snail abundance was required for clonogenic capacity of starved cancer cells (Fig. 1b). However, knockdown of Snail after starvation did not affect the clonogenic potential, indicating that the protective effect of Snail is important in the early response to glucose deprivation (Supplementary Fig. 1d). Another EMT inducer, Slug (SNAI2), functioned similarly to Snail in terms of cell death and clonogenic potential under metabolic stress (Supplementary Fig. 1e). Conversely, gain of Snail potentiated cancer cell survival and clonogenic capacity, overcoming glucose starvation or paclitaxel treatment (Supplementary Fig. 1f–h). Matrix detachment of carcinoma cells, which cancer cells inevitably encounter during metastatic progression, also leads to profound metabolic stress due to loss of glucose transport[3]. Examining metabolic stress induced by anchorage loss, we found that Snail abundance was also required for cell survival following matrix detachment (Fig. 1c). *In vivo*, the matrix-detached cancer cells must survive before tumour growth in orthotopically implanted tissue as well as metastatic progression via intravascular circulation. Consistent with the well-established role of Snail in cancer progression[13,14,18], lack of Snail largely attenuated *in vivo* tumour initiation of orthotopically injected MDA-MB-231 cells without extracellular matrix (Supplementary Fig. 2a). When these cells were injected into tail vein, Snail knockdown significantly suppressed the metastatic potential while its overexpression increased metastatic capacity (Fig. 1d and Supplementary Fig. 2b). These results support that Snail abundance is prerequisites for cancer cell survival before *in vivo* tumour initiation and metastatic progression of cancer cells.

Because oxidative stress rather than ATP depletion is the primary trigger of cell death under metabolic stress[3,19,20], we next measured reactive oxygen species (ROS) and NADPH level following Snail knockdown to investigate the metabolic link between Snail abundance and cell death. Loss of Snail significantly increased ROS level in breast cancer cells, especially in glucose-deprived condition (Fig. 1e), and antioxidant treatment reversed cell death of Snail knockdown cells under starvation (Fig. 1f and Supplementary Fig. 2c). Because NADPH is the major reduction power against oxidative stress, we next measured cellular NADPH level according to Snail abundance. Indeed, loss of Snail decreased NADPH level (Fig. 1g), while gain of Snail function consistently increased NADPH level in a glucose-6-phosphate dehydrogenase (G6PD)-dependent manner (Fig. 1h and Supplementary Fig. 2d), suggesting that Snail regulates NADPH homeostasis via PPP and thereby confers cancer cell survival against oxidative stress.

**Snail suppresses glycolytic activity in breast cancer cells**. To gain more insight into Snail function on glucose metabolism, we performed metabolic profiling in terms of Snail abundance using [1]H-NMR spectroscopy. A total of 22 metabolites were identified in MDA-MB-231 breast cancer cell extracts using targeted metabolic profiling. Orthogonal partial least-squares discriminant analysis (OPLS-DA) was applied to maximize intergroup variance. In the OPLS-DA score plot, breast cancer cells lacking Snail were clearly separated from those of controls (Fig. 2a). Fourteen metabolites were found to significantly distinguish the loss of Snail and control group (Fig. 2a). Similar results were observed in untargeted metabolic profiling of MDA-MB-231 cells (Supplementary Fig. 3a,b). Intriguingly, loss of Snail increased endogenous intermediate metabolites of aerobic glycolysis and lactate, a hallmark end product of glycolysis (Fig. 2b,c). Amino acid biosynthesis, including of serine-driven one carbon units such as glycine, was increased by loss of Snail compared with control. The inverse relationship

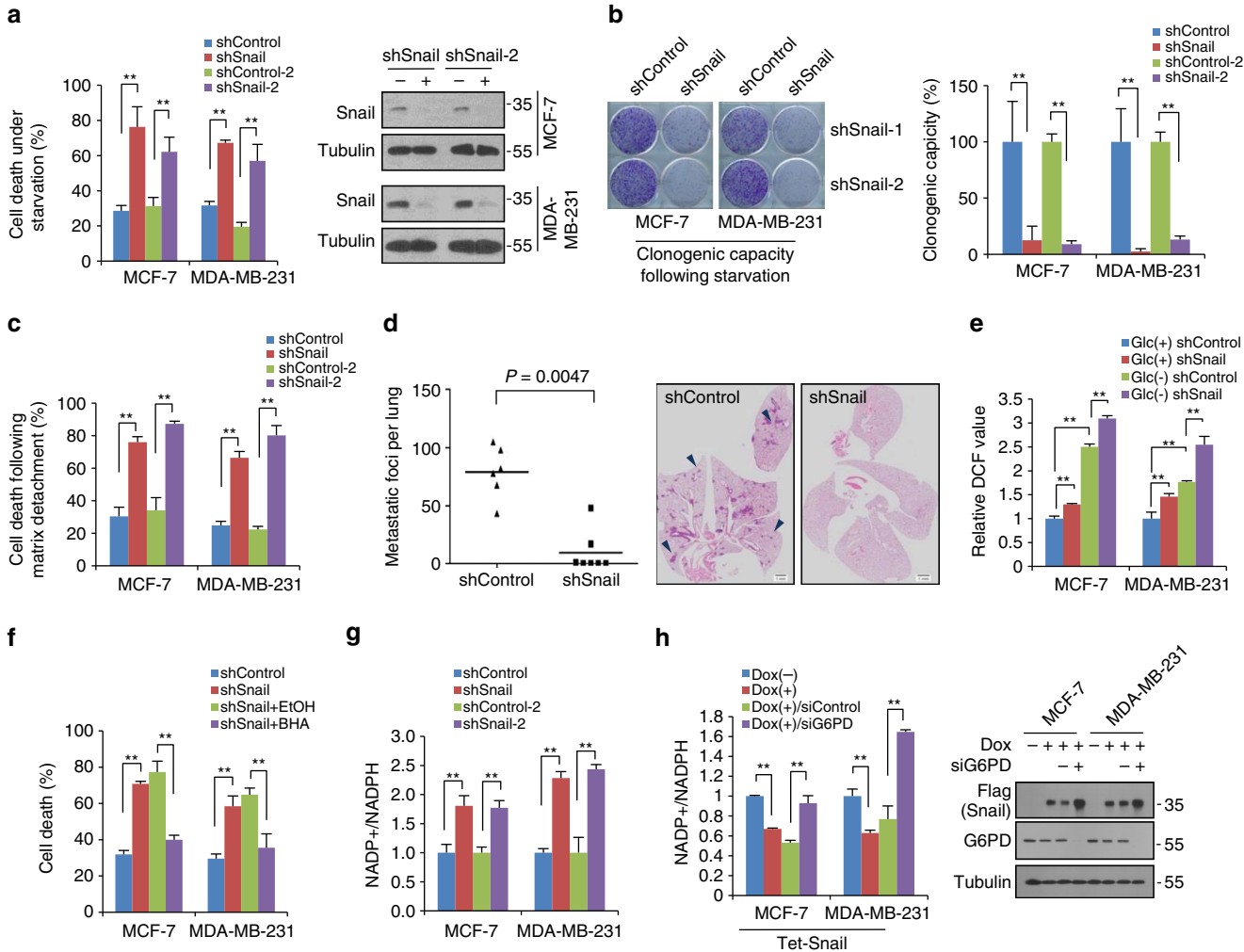

**Figure 1 | Snail potentiates cancer cell survival under starvation via overcoming oxidative stress.** (**a**) Cells were cultured in absence of glucose, and cell death was quantitated by trypan blue exclusion assay (left). Immunoblot analysis shows endogenous Snail protein abundance (right). In total, 20 μg of cell lysates were used to detect endogenous Snail, and loading controls were validated from reprobing the same blot. Antibody validations for immunoblot analysis are provided in Supplementary Fig. 10. (**b**) Clonogenic survival assay of cancer cells following glucose starvation as described in 'Methods' section (left). Colonies of more than 50 cells were counted after crystal violet staining (right). (**c**) Cell death of breast cancer cells expressing control-shRNA or Snail-shRNA cultured 24 h after plating in attached or detached (poly-HEMA-coated) plates. (**d**) Lung metastasis by tail vein xenograft of MDA-MB-231-D3H2LN cells. In total, $5 \times 10^5$ cells either of control (shControl, $n = 6$) or of knockdown of Snail (shSnail, $n = 7$) were inoculated intravenously into immunodeficient mice. The number of lung metastatic nodules at day 28 was counted under microscopic examination (left). Statistical significance was determined by Mann–Whitney test. Whole-field images of representative lungs that showed median value for each group (right). Arrows indicate metastatic tumour foci in mouse lung. Scale bar, 1 mm. (**e**) The cancer cells expressing control-shRNA or Snail-shRNA were incubated in the presence of glucose (Glc + ) or absence (Glc − ) for 2 h. The ROS levels are expressed as the relative change in the mean DCF values relative to the glucose-treated control shRNA. (**f**) Antioxidant BHA (100 μM) treatment rescued cell death induced by glucose deprivation of Snail knock-downed cells. (**g**) The effect of Snail knockdown on NADP$^+$/NADPH ratio in cancer cells. (**h**) Inducible Snail overexpression increased NADPH production in a G6PD-dependent manner. Snail was induced by treatment of doxycycline (Dox) for 48 h in combination with control or G6PD siRNA. NADP$^+$/NADPH ratio (left) and protein abundance (right) were determined. In total, 5 μg of cell lysates were used to detect overexpressed flag-tagged Snail and endogenous G6PD, and loading controls were validated from reprobing the same blot. Data are means ± s.d. from $n = 3$ (**a**–**c**) or $n = 5$ (**e**–**h**) independent experiments. Statistical significances compared with control were denoted as *$P < 0.05$; **$P < 0.01$ by a two-tailed Student's $t$-test. Unprocessed original scans of blots are shown in Supplementary Fig. 12.

between Snail abundance and lactate production was further confirmed by independent experiments on Snail loss or gain (Fig. 2d and Supplementary Fig. 3c). Consistently, we successfully traced the metabolic fate of $^{13}$C-glucose to $^{13}$C-lactate following knockdown of Snail in MDA-MB-231 cells (Fig. 2e and Supplementary Fig. 3d). These results indicate that Snail leads to metabolic reprogramming via repression of glycolytic flux in breast cancer cells.

**Snail regulates phosphofructokinase PFKP expression.** To elucidate the mechanistic link between Snail and metabolic reprogramming, we postulated that transcriptional repressor Snail directly regulates the rate-limiting step of aerobic glycolysis. Previous microarray studies have shown that many genes involved in glycolysis were repressed by Snail[21,22]. In terms of Snail function on glycolytic activity, we focused on PFK-1, which catalyses the first committed step of glycolysis, irreversibly

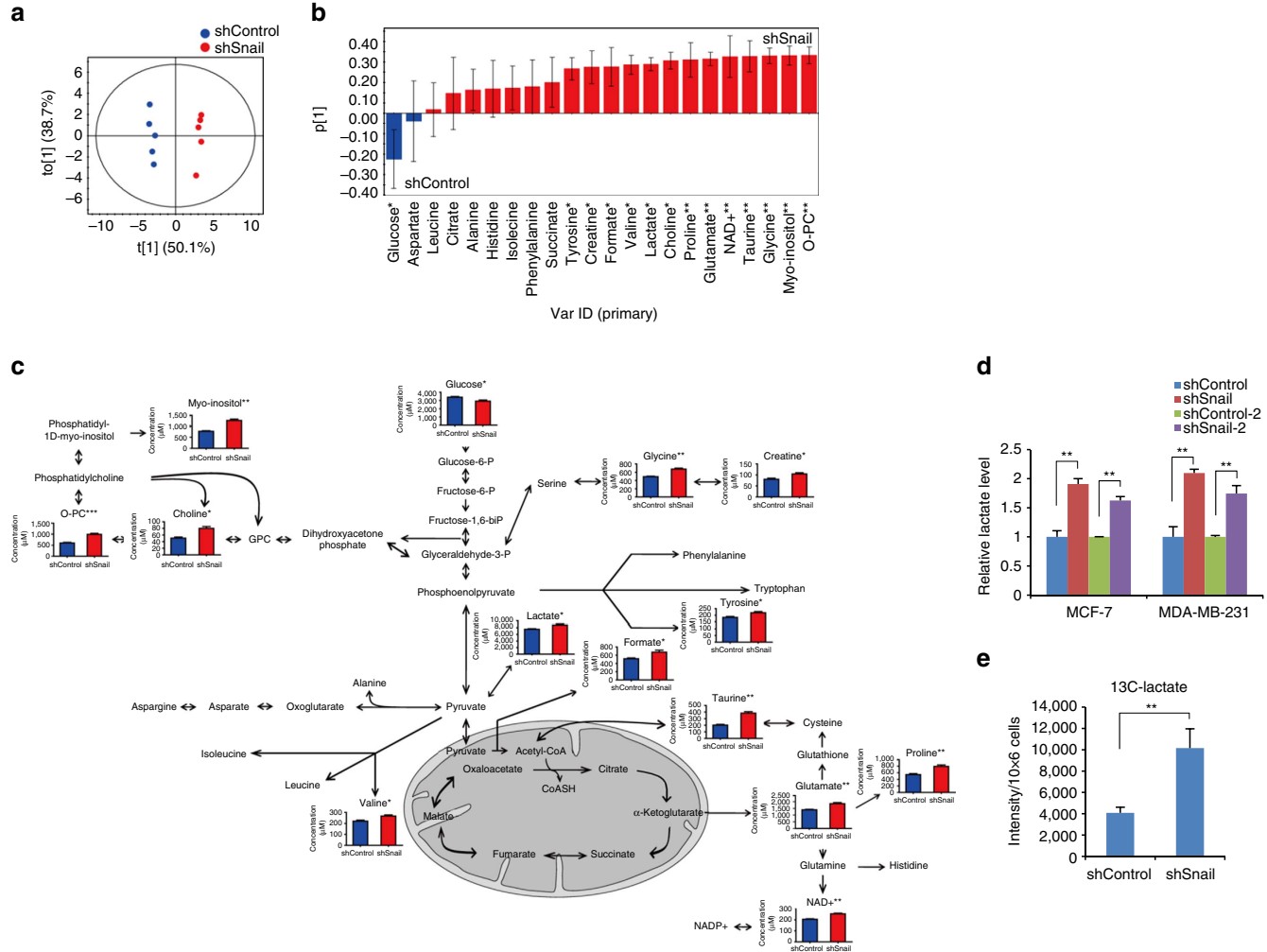

**Figure 2 | Snail suppresses aerobic glycolysis in cancer cells. (a,b)** Metabolic profile analysis following knockdown of Snail in MDA-MB-231 cells. OPLS-DA score plot ($R^2X = 0.888$, $R^2Y = 0.992$, $Q^2 = 0.945$) (**a**) and loading column plot (**b**) of metabolite concentrations through targeted profiling derived from [1]H NMR spectra of MDA-MB-231 cell extracts expressing control shRNA (shControl) and Snail shRNA (shSnail). (**c**) A diagram and quantification of the changes in intracellular metabolites in MDA-MB-231 cells. *,** and *** indicate $P < 0.05$, $P < 0.01$ and $P < 0.001$ calculated from the Mann–Whitney U test. NAD +, nicotinamide adenine dinucleotide (oxidized form); O-PC, O-phosphocholine. The x and y axis represent experimental groups (blue bars, shControl; red bars, shSnail) and metabolite concentration (μM), respectively. (**d**) Quantification of intracellular lactate following knockdown of Snail in MCF-7 and MDA-MB-231 cells. ** $P < 0.01$ compared with the control by Student's t-test. (**e**) The relative abundance of [13]C-lactate from MDA-MB-231 cell extracts was quantified from 1D [1]H{[13]C} HSQC peak intensity. Data are means ± s.d. from $n = 5$ independent experiments. ** $P < 0.01$ compared with the control by Student's t-test.

converting F6P to F1,6P. Because PFK-1 functions as the gatekeeper of glycolytic flux, its expression and activity are tightly regulated[10]. Similar to pyruvate kinase[23], three isoforms of PFK-1 exist from mammalian: PFKL and PFKM, expressed in liver and muscle, respectively, and PFKP, mainly expressed in platelet and cancer[24,25]. When we re-evaluated the abundance of PFK-1 isoforms, the PFKP was preferentially detected in human cancer cells while non-tumourigenic MCF-10A cells mainly expressed PFKL (Fig. 3a and Supplementary Fig. 4a, validation of cell line in Supplementary Fig. 11). Thus, PFKP isoform is a major PFK-1 controlling the first rate-limiting step of aerobic glycolysis in human cancer cells. Importantly, the transcripts, protein abundance and kinase activity of PFKP were consistently increased by loss of Snail (Fig. 3b and Supplementary Fig. 4b), while they were suppressed by Snail overexpression (Supplementary Fig. 4c,d). Because posttranslational modification of PFK-1 results in inhibition of the kinase activity[26], we further determined whether Snail abundance modulates O-glycosylation

or phosphorylation of PFKP. However, post-translational modifications were not affected by Snail (Supplementary Fig. 4e), indicating that Snail represses PFKP mainly on a transcriptional level, similar to E-cadherin and BRCA-1 (refs 27,28). Indeed, as the PFKP proximal promoter harbours multiple Snail-binding canonical E-boxes (CACCTG), we could clone the promoter region of PFKP into the reporter vector with mutations of the putative Snail binding sites individually or in combination (Fig. 3c). The PFKP reporter activity increased when we silenced Snail, whereas the specific mutation of predicted binding sites attenuated the ability of Snail to suppress reporter activity (Fig. 3d). We next used chromatin immunoprecipitation (ChIP) assay to determine whether Snail comprises a repressor complex in PFKP promoter. As expected, a DNA fragment containing the E-boxes can be amplified from the genomic DNA immunoprecipitated samples from antibody of endogenous Snail in an E-box dependent manner (Fig. 3e and Supplementary Fig. 4f).

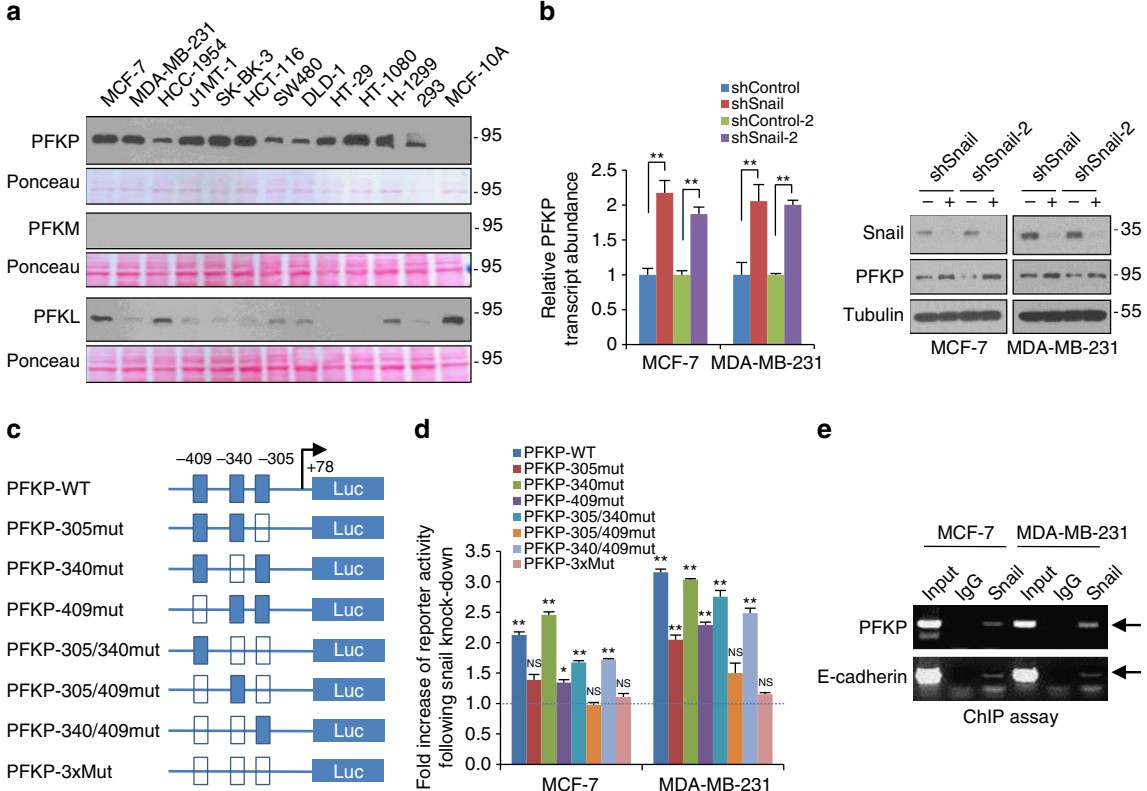

**Figure 3 | PFKP is a major isoform of PFK-1 in cancer cells and a target of Snail repressor.** (**a**) PFK-1 immunoblotting of cell lysates from various human cancer cells. In total, 5 μg of each cell lysate for PFKP and 30 μg of cell lysate for PFKM or PFKL were subjected to immunoblot analysis. The films were developed at the same exposures to compare relative protein abundance of PFK-1 isoforms, and protein loading of each blot was validated with Ponceau stain. The PFK-1 antibody validation using positive control of platelet, muscle and liver tissue are shown in Supplementary Fig. 10c. Cell lines used in this study was authenticated by short tandem repeat profiling as shown Supplementary Fig. 11. (**b**) Relative transcript (left) and protein (right) abundance of PFKP following knockdown of Snail (shSnail) in breast cancer cells. A total of 20 μg and 5 μg of cell lysates were used to detect endogenous Snail and PFKP, respectively. Representative blots are shown from at least two independent experiments (**a,b**). (**c**) Schematic diagram showing positions of potential Snail-binding canonical E-boxes on the PFKP proximal promoter reporter constructs. Arrow, transcription start site; empty boxes denote E-box mutant. (**d**) Fold increase of reporter activities in combination with wild type or mutated PFKP promoter following Snail shRNA compared with each control shRNA in breast cancer cells. *$P < 0.05$; **$P < 0.01$ compared with the control. (**e**) Chip-enriched DNA was determined by RT-PCR using specific primers complementary to the promoter regions containing E-box of PFKP (upper) and E-cadherin (lower). Statistical significances (**b,d**) compared with control were denoted as *$P < 0.05$; **$P < 0.01$ by a two-tailed Student's $t$-test. Unprocessed original scans of blots are shown in Supplementary Fig. 12.

**PFKP is a key flux controller between glycolysis and PPP.** We next examined the regulatory role of PFKP in metabolic reprogramming and cancer cell survival. An OPLS-DA score plot of targeted metabolic profiles from $^1$H NMR spectra of MDA-MB-231 cells revealed remarkably different metabolic patterns by inducible knock-down of PFKP (Fig. 4a). Due to the first rate-limiting role of PFK-1 on glycolysis, knockdown of PFKP significantly suppressed lactate production and amino acids biosynthesis from phosphoenolpyruvate and pyruvate (Fig. 4b–d). Similar results were observed through untargeted metabolic profiling following inducible knockdown of PFKP in MDA-MB-231 cells (Supplementary Fig. 5a,b). Tracing with $^{13}$C-glucose further showed suppression of $^{13}$C-lactate according to the PFKP knockdown (Fig. 4e and Supplementary Fig. 5c). Consistent with previous findings that oxidative phosphorylation was independent on glycolytic flux[29], the mitochondrial oxygen consumption rate in either the presence or absence of oligomycin was slightly decreased by suppression of PFKP (Fig. 4f and Supplementary Fig. 5d). These results demonstrate that (1) PFKP is a gate-controller of glycolytic flux in cancer cells, and (2) PFKP functions inversely to Snail on glycolytic activity.

Recent findings suggest that inhibition of glycolysis redirects glucose flux towards the PPP[19,26,30]. To examine the role of PFKP

in regulating glucose flux between glycolysis and PPP, we next determined NADPH and ROS levels in breast cancer cells according to the PFKP abundance. Loss of PFKP increased NADPH level in a G6PD-dependent manner (Fig. 5a and Supplementary Fig. 6a,b). Because the serine-driven one carbon pathway in glycolysis can provide a large fraction of NADPH[31], we next performed mass spectrometry analysis to quantitatively measure endogenous metabolites of oxidative PPP. Indeed, suppression of PFKP increased the amount of ribulose-5-phosphate (R5P; Fig. 5b), supporting that PFKP regulates glucose flux into PPP in cancer cells. Since PFK-1 deficiency leads to glycogen storage disease (known as Tarui's disease) via conversion of glucose-6-phosphate to glucose-1-phosphate[32], we next examined glycogen amounts to determine whether PFKP silencing leads glucose reflux to glycogenesis in cancer cells. However, the glycogen level was not increased by PFKP silencing regardless of glucose concentration in breast cancer cells (Supplementary Fig. 6c).

Examining the functional relevance of metabolic reprogramming by suppression of PFKP, we found that suppression of glycolytic activity coupled with increased PPP flux led the reversible G0-G1 arrest similar to cancer dormancy (Supplementary Fig. 6d,e)[17,33]. Importantly, suppression of PFKP significantly

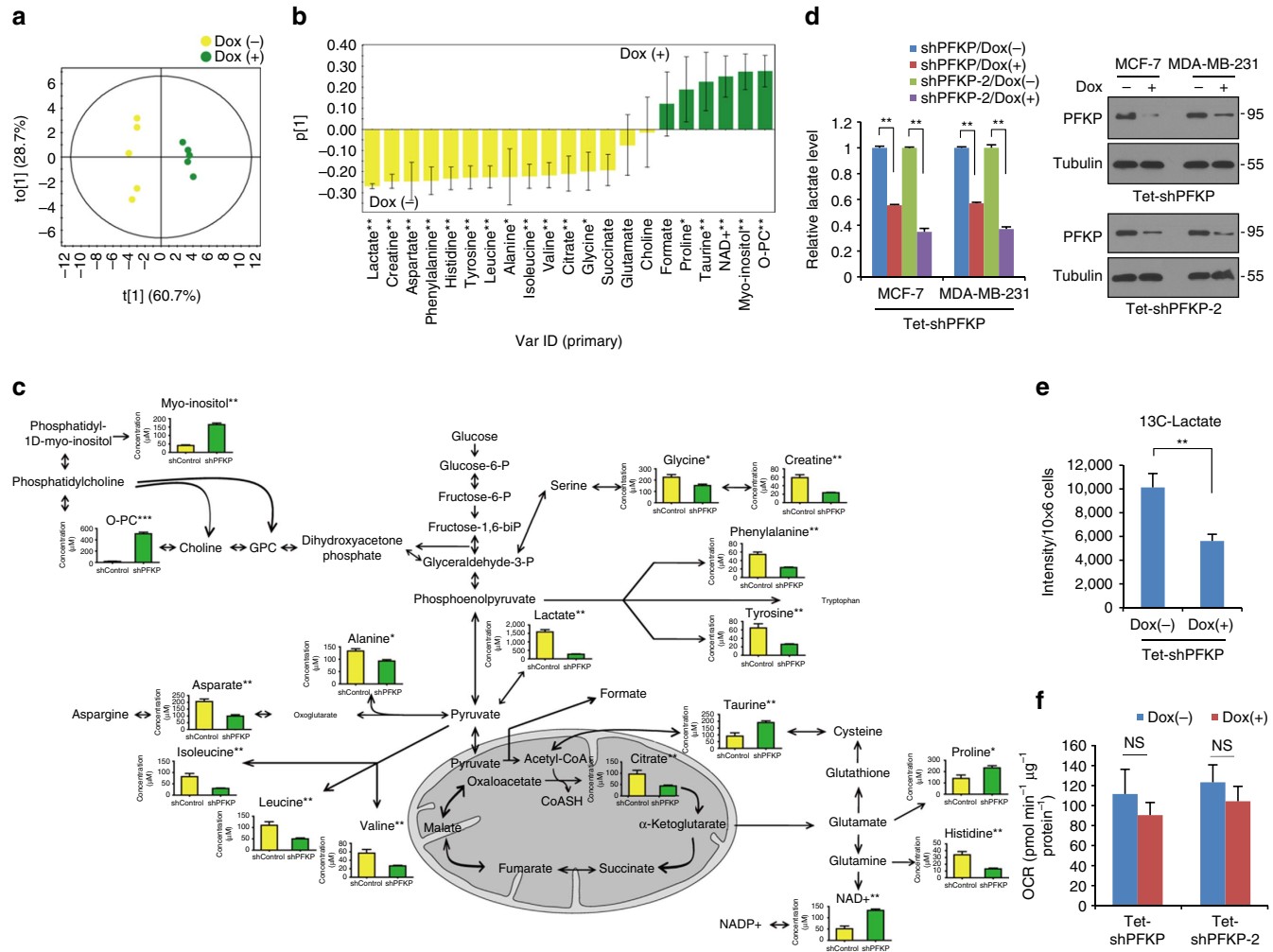

**Figure 4 | PFKP is the gate controller of glucose flux into glycolysis.** (**a,b**) Metabolic profile of NMR spectra following inducible knockdown of PFKP in MDA-MB-231 cells. OPLS-DA score plot ($R^2X = 0.893$, $R^2Y = 0.985$, $Q^2 = 0.977$) (**a**) and loading column plot (**b**) of metabolite concentrations through targeted profiling derived from $^1H$ NMR spectra of MDA-MB-231 cell extracts treated with control shRNA (Dox −) and PFKP shRNA (Dox + ). (**c**) A diagram and quantification of the changes in intracellular metabolites following inducible knockdown of PFKP in MDA-MB-231 cells. *,** and *** indicates $P < 0.05$, $P < 0.01$ and $P < 0.001$ calculated from Mann–Whitney U test. NAD +, nicotinamide adenine dinucleotide (oxidized form); O-PC, O-phosphocholine. The x and y axis represent experimental groups (yellow bars, shControl; green bars, shPFKP) and metabolite concentration (μM), respectively. (**d**) Relative lactate level following doxycycline-inducible PFKP shRNA (Dox + ) in breast cancer cells (left). Immunoblot analysis shows PFKP protein abundance in two independent sets of shRNA against PFKP (right). In total, 5 μg of cell lysates were used to detect endogenous PFKP, and loading controls were validated from reprobing the same blot. The image is representative of two independent experiments. (**e**) The relative abundance of $^{13}C$-lactate from the $^{13}C$-glucose tracer experiment in MDA-MB-231 cells. (**f**) Mitochondrial OCR (delta OCR pre- and post- rotenone) in Tet-inducible PFKP shRNA. Data are means ± s.d. from n = 3 (**d**) or n = 5 (**e,f**) independent experiments. Statistical significances compared with control are denoted as *$P < 0.05$; **$P < 0.01$ by a two-tailed Student's t-test. Experiment (**d**) is representative and was replicated from at least two independent experiments.

potentiated cancer cell survival and clonogenic capacity under metabolic stress (Fig. 5c,d), and RNAi-resistant PFKP expression vector could successfully attenuate the survival and clonogenic potential of cancer cells induced by shRNA-mediated PFKP knockdown (Supplementary Fig. 6f,g). Because anti-oxidative capacity is closely related to chemotherapeutic resistance of cancer cells[34], we next examined the role of PFKP suppression on cancer cell survival against paclitaxel treatment. Intriguingly, inducible knockdown of PFKP significantly increased clonogenic capacity of breast cancer cells against paclitaxel treatment (Fig. 5e), suggesting that suppression of PFKP followed by redirection of glucose flux towards PPP plays an important role in slowing cell growth coupled with therapeutic resistance of cancer cells.

The maintenance of PPP flux is responsible for cancer cell survival against oxidative stress induced by matrix detachment[3,19,26]. To form metastatic colonization as well as to gain tumour-initiation capability, matrix-detached carcinoma cells must survive before proliferative outgrowth in the foreign microenvironment[1,18]. Therefore, our observations suggest a possibility that metabolic reprogramming towards PPP via suppression of PFKP can increase the tumour-initiating and metastatic potential of cancer cells by overcoming metabolic stress. To examine whether metabolic reprogramming regulated by PFKP plays a role in tumour- and metastatic-initiation[18,35], we designed an *in vivo* experiment controlling PFKP abundance with an inducible system at the initial period of tumour implantation or systemic circulation (Fig. 5f). Interestingly, transient loss of

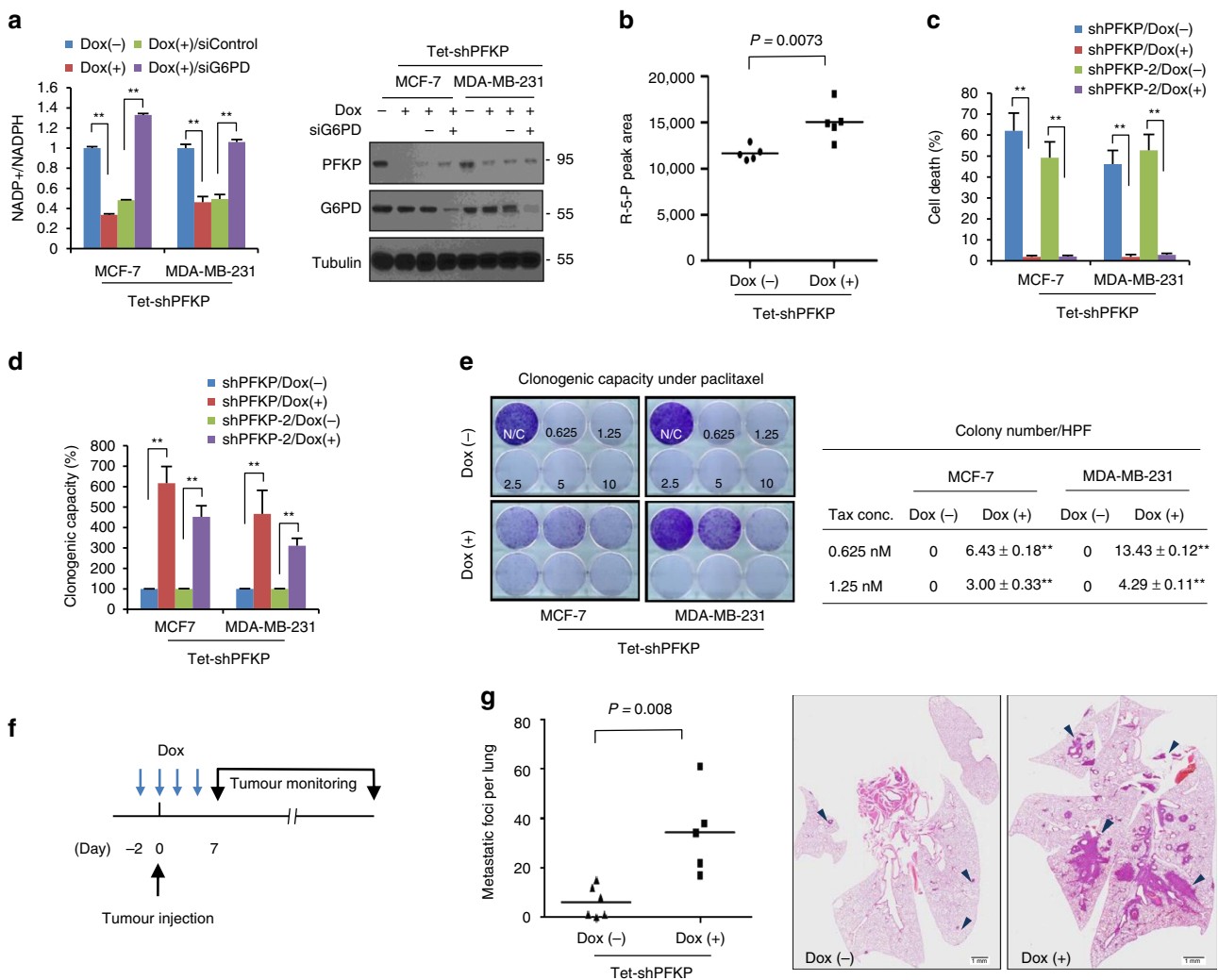

**Figure 5 | Repression of PFKP induces glucose reflux towards PPP thus providing survival advantage to breast cancer cells.** (**a**) PFKP controls NADPH level via G6PD-dependent manner. The NADP$^+$/NADPH ratio (left) and protein abundance (right) of indicated cells were measured following inducible knockdown of PFKP (Dox + ) in combination with control siRNA or G6PD siRNA. In total, 5 µg of cell lysates were used to detect PFKP and G6PD, and loading controls were validated from reprobing the same blot. (**b**) Quantitative mass spectrometry analysis of PPP metabolite following inducible knockdown of PFKP. HPLC-ESI-MS/MS peak areas of ribulose-5-phosphate/ribose-5-phosphate of cells expressing PFKP shRNA (Dox + ) compared with control (Dox − ). (**c,d**) Cell death quantification (**c**) and clonogenic capacity (**d**) after glucose starvation following inducible knockdown of PFKP (Dox + ) compared with control (Dox − ). (**e**) Clonogenic survival of breast cancer cells against paclitaxel treatment as indicated by concentration. The PFKP was knocked-down with doxycycline (Dox + ) for 48 h before paclitaxel treatment (left). The colony number was determined by stereomicroscopic examination under high power field (right). Data are means ± s.d. from $n = 3$ (**a**) or $n = 5$ (**c–e**) independent experiments. Statistical significances compared with control are denoted as *$P < 0.05$; **$P < 0.01$ by a two-tailed Student's $t$-test. (**f**) An experimental design to determine functional relevance of dynamic regulation of PFKP. Inducible PFKP shRNA expressing cells were either untreated (Dox − ) or pretreated with doxycycline (Dox + ) in vitro 48 h before inoculation. Recipient mice received either PBS (Dox − ) or doxycycline via intraperitoneal delivery at day 0, 2, 4. (**g**) Lung metastasis by tail vein xenograft of MDA-MB-231-D3H2LN cells. $1 \times 10^5$ cells either of control (Dox − , $n = 6$) or of transient knockdown of PFKP (Dox + , $n = 5$) were inoculated intravenously into immunodeficient mice. The number of lung metastatic nodules was counted under microscopic examination (left). Statistical significance was determined by Mann–Whitney test. Whole-field images of representative lungs that showed median metastatic value for each group (right). Arrows indicate metastatic tumour foci in mouse lung. Scale bar, 1 mm.

PFKP during an initial period in quantities that generated suboptimal engraftment significantly enhanced *in vivo* tumour initiation (Supplementary Fig. 6h). Further, the dynamic suppression of PFKP increased the lung metastatic potential of MDA-MB-231 cells (Fig. 5g). These results reveal that dynamic repression of PFKP plays an important role in potentiating cancer cell survival before *in vivo* tumour initiation and metastatic progression.

During the EMT program, cancer cells acquired metastatic potential accompanied with suppression of epithelial genes and increased abundance of mesenchymal genes. To assess whether PFKP constitutes a mediator of Snail-mediated phenotypic conversion, we next examined transcript abundance of representative EMT phenotypic genes according to the PFKP status. Interestingly, PFKP abundance did not affect the transcript abundance of EMT genes which were regulated by Snail in breast cancer cells (Supplementary Fig. 7), indicating that the Snail-PFKP axis mainly involves glucose metabolism independent of phenotypic conversion. Canonical Wnt signalling is a well-known upstream regulator of Snail controlling cancer progression, while

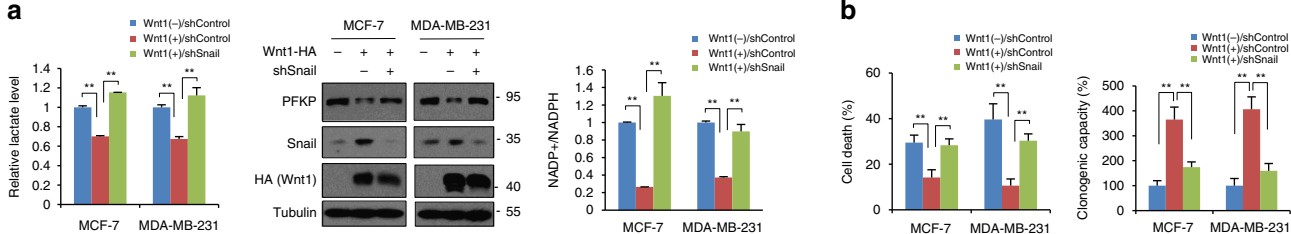

**Figure 6 | Metabolic reprograming by Wnt1 potentiates cancer cell survival in a Snail-dependent manner.** (**a**) Relative lactate production (left), immunoblot analysis (middle) and NADP$^+$/NADPH ratio in breast cancer cells transduced with Wnt1 expressing vector (Wnt1 + ). Snail was silenced with shRNA control (shControl) or with Snail shRNA (shSnail). Data are means ± s.d. from $n = 3$ independent experiments of biological replicates. Twenty for Snail and 5 µg for PFKP and HA-tagged Wnt1 of cell lysates were used. (**b**) Cell death under glucose starvation (left) and clonogenic capacity of the starved cells (right). Data are means ± s.d. from $n = 5$ independent experiments. ** denote $P < 0.01$ by Student's $t$-test. Images are representative of two independent experiments. Unprocessed original scans of blots are shown in Supplementary Fig. 12.

metabolic reprogramming by Wnt is not well-known[13,36,37]. Thus, our observations raised the question of whether the canonical Wnt pathway is involved in metabolic reprogramming in a Snail-dependent manner. We found that induction of Wnt1 increased NADPH production, cancer cell survival and clonogenic capacity under metabolic stress while knockdown of Snail rescued Wnt-induced metabolic reprogramming (Fig. 6). These results provide novel observations that the Snail plays a role, at least in part, in metabolic reprogramming regulated by canonical Wnt pathway.

**Roles of PFKP during metabolic reprogramming by Snail.** Finally, given the ability of Snail to potentiate cell survival with an inverse metabolic rewiring to PFKP, we sought to determine whether PFKP constitutes a key element in the reprogramming of glucose metabolism by Snail. To confirm the Snail-PFKP axis, we designed an experimental system in which modulation of PFKP could rescue metabolic reprogramming and cancer cell survival regulated by Snail. Indeed, the lactate and NADPH production following loss of Snail were largely relieved by inducible knock-down of PFKP (Fig. 7a,b and Supplementary Fig. 8a). The regulatory axis was also functional on cell death and clonogenic capability under glucose starvation or induced by matrix detachment (Fig. 7c–e and Supplementary Fig. 8b). Further, dynamic suppression of PFKP restored the *in vivo* tumour-initiating and metastatic potential of cancer cells having a Snail-knockdown background (Fig. 6f and Supplementary Fig. 8c). Overexpression of PFKP consistently rescued Snail-mediated metabolic reprograming by means of increased glycolytic activity, decreasing NADPH level and cancer cell survival (Supplementary Fig. 9a,b). Further, the overexpression of PFKP attenuated tumour-initiation and metastatic potential *in vivo* (Supplementary Fig. 9c,d). Hence, PFKP plays a key role in regulating Snail-mediated metabolic reprogramming in breast cancer cells (Fig. 6g).

**Discussion**

In aerobic glycolysis, the reactions catalysed by hexokinase, PFK-1 and pyruvate kinase (PK) are accompanied by high free-energy change and are virtually irreversible[38]. Therefore metabolic reprogramming of cancer cells mainly regulates these irreversible enzymatic steps for efficient proliferation or survival depending on microenvironment resources. Indeed, a specific isoform of PK is tightly regulated by receptor tyrosine kinase and ROS level[19,23]. As a transcriptional repressor, Snail can regulate many genes involved in glucose metabolism, such as glucose phosphate isomerase and aldolases[21], suggesting that Snail participates in complex metabolic reprogramming in cancer

cells. This study has aimed to prove the hypothesis that the Snail-PFKP axis plays an important regulatory role in aerobic glycolysis. First, lactate production by Snail-gain or –loss consistently supports the notion that Snail suppresses aerobic glycolysis. Second, global profiling of endogenous metabolites as well as tracing analysis with $^{13}$C-glucose showed Snail and PFKP to be inversely correlated. Third, knock-down of PFKP largely rescued metabolic reprogramming and cell death induced by Snail-loss, supporting that Snail regulates glycolytic activity via gatekeeper PFKP. In this light, recent findings regarding loss of fructose-1,6-bisphosphatase 1 (FBP1) in triple negative breast cancer and renal cell carcinoma are of considerable interest[24,25], because FBP1 is involved in gluconeogenesis by means of reverse reaction catalysed by PFK-1. Considering glucose produced from gluconeogenesis is mainly consumed by the brain under starvation and glycolysis[38], suppression of FBP1 in cancer cells can also provide survival advantage under metabolic stress, such as hypoxic or glucose-starved condition[24]. Thus, bidirectional suppression of PFKP and FBP1 by Snail allow tight glucose flux control towards PPP and efficient glycolysis, providing survival advantage and enabling a critical trade-off between cell proliferation and survival during cancer progression[23,24,39].

We observed that the PFKP isoform prevails over PFKM or PFKL in human cancer cells[25,40]. Interestingly, compared with other isoforms, PFKP is less sensitive to feedback inhibition by ATP and more sensitive to activation by fructose 2,6-bisphosphate (F2,6BP), the most potent allosteric activator of PFK-1 (refs 41,42). While we showed that Snail suppressed PFKP activity at the transcriptional level, it should be noted that oncogenic regulations of PFK-1 kinase activity have been reported. In response to hypoxia, posttranslational *O*-linked β-*N*-acetylglucosamine (*O*-GlcNAc) modification of PFK-1 resulted in inhibition of the kinase activity and redirected glucose flux towards PPP[26]. Intriguingly, *O*-GlcNAc modification also stabilizes Snail protein by inhibition of GSK-3-mediated phosphorylation and subsequent degradation[43]. These indicate that *O*-GlcNAc modification on PFK-1 and Snail synergistically controls glucose flux towards PPP. In turn, uridine diphospho-N-acetylglucosamine (UDP-GlcNAc) biosynthesized from glucose, glutamine, acetyl-co-A and ATP may serve as a regulator of glucose flux via posttranslational modification of Snail and PFK-1 (ref. 26). The p53 tumour-suppressor induced TIGAR lowered F2,6BP in cells, resulting in inhibition of PFK-1 and glycolytic activity[9]. Conversely, loss of p53 function in human cancer provides active PFK-1 by increased glucose uptake and F2,6BP level, allowing high glycolytic activity for biomass synthesis[9,11].

The reducing power of NADPH is critically required not only for macromolecule synthesis during cancer cell proliferation, but

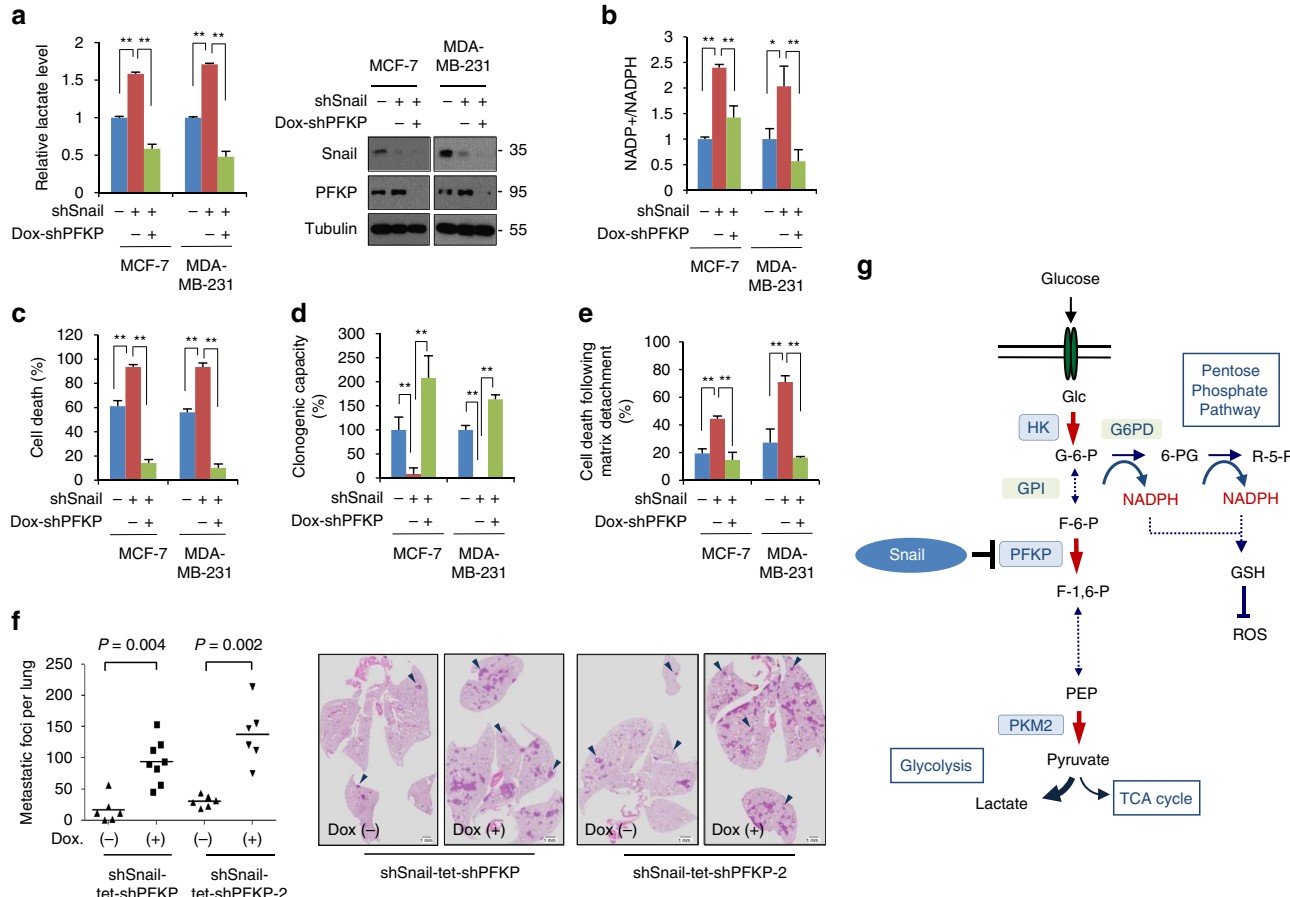

**Figure 7 | Snail-PFKP axis regulating glucose flux between glycolysis and PPP.** (**a**) Relative lactate production (left) and immunoblot (right) in breast cancer cells transduced with shRNA control (shSnail −) or with Snail shRNA (shSnail +). PFKP was knocked down by treatment with doxycycline (Dox +) for 24 h in Snail shRNA cells. A total of 20 μg and 5 μg of cell lysates were used to detect Snail and PFKP, respectively. (**b–e**) Inducible knockdown of PFKP rescued lack of Snail in breast cancer cells. The NADP$^+$/NADPH ratio (**b**), cell death (**c**), and clonogenic capacity (**d**) following glucose starvation and cell death after matrix detachment (**e**) were measured. Data are means ± s.d. from $n = 3$ (**a–e**) independent experiments. Statistical significances compared with control were denoted as *$P < 0.05$; **$P < 0.01$ by a two-tailed Student's $t$-test. (**f**) Lung metastasis by tail vein xenograft of Snail-expressing MDA-MB-231-D3H2LN cells. In total, $5 \times 10^5$ cells either of control (Dox −, $n = 6$) or of transient knockdown of PFKP (Dox +, $n = 8$ for shPFKP and $n = 6$ for shPFKP-2) were inoculated intravenously into immunodeficient mice. Inducible PFKP shRNA expressing cells were either untreated (Dox −) or treated with doxycycline (Dox +) as described in Fig. 5f. The number of lung metastatic nodules at day 28 was counted under microscopic examination (left). Statistical significance was determined by Mann–Whitney test. Whole-field images of representative lungs that showed median metastatic value for each group (middle and right). Arrows indicate metastatic tumour foci in mouse lung. Scale bar, 1 mm. (**g**) A schematic diagram depicting a potential mechanism by which the Snail/PFKP axis regulates glucose flux towards PPP in cancer cells. Red arrows denote irreversible rate-limiting steps in glycolysis. G-6-P, glucose-6-phosphate; F-6-P, fructose-6-phosphate; F-1,6-P, fructose-1,6-bisphosphate; PEP, phosphoenolpyruvate; 6-PG, 6-phosphogluconate; R-5-P, ribulose-5-phosphate; HK, hexokinase; GPI, glucose phosphate isomerase; G6PD, glucose-6-phosphate dehydrogenase; PKM2, pyruvate kinase M2 isoform.

also to resist oxidative stress under starved condition[3–5]. Indeed, the ROS detoxification capability is more important than anabolic biomass production in advanced cancer[3,44]. Interestingly, p53 suppresses PPP and NADPH production via binding to G6PD, and p53 mutants lack the G6PD-inhibitory function resulting in enhanced PPP flux in human cancer[11]. In this study, we observed that the canonical Wnt pathway is also involved in NADPH homeostasis via Snail-mediated PFKP suppression. However, it should be noted that nuclear factor-like 2, a transcription factor regulating NADPH homeostasis, is also regulated by GSK-3 and the β-TrCP degradation complex, similar to Snail and beta-catenin[36,37,45,46].

While transcriptional repressor Snail has long been noted as a critical phenotypic EMT inducer[27], recent findings suggest that EMT comprises a wide range of cancer cell biologic behaviour, such as cancer stemness and therapeutic resistance[16]. Interestingly, the PFKP abundance was independent to phenotypic

changes of EMT markers. Therefore, our and recent observations indicate that regulations of glycolysis and gluconeogenesis implicate a wide range of cellular functions of Snail repressor beyond phenotypic conversion during the cancer progression[24]. Our findings may also provide a mechanistic link accounting for previous observations that cancer stem cells or cancer cells undergoing EMT are resistant to cell death accompanied by increased detoxification of ROS and slow cell proliferation[17,44,47]. Thus, Snail-PFKP axis regulating glucose flux into PPP and NADPH homeostasis, especially in a resource-limited catabolic environment, constitutes an important metabolic programming during the solid cancer progression[4,8,11].

## Methods

**Cell culture conditions and western blotting.** MDA-MB-231 cells (a gift from G. Mills) and HCC-1954 cells were cultured in a RPMI1640 with 5% FBS (HiClone). MCF-10A cells (a gift from M. Wicha) were cultured in DMEM/F12

with 5% horse serum, 20 µg ml$^{-1}$ EGF, 0.5 µg ml$^{-1}$ hydrocortisone, 0.1 µg ml$^{-1}$ cholera toxin, 5 µg ml$^{-1}$ insulin and 100 IU ml$^{-1}$ penicillin/streptomycin. Metastatic MDA-MB-231-D3H2LN cells (Caliper LifeSciences) were maintained with Eagle's MEM containing 10% FBS. All other cells were cultured in DMEM (Invitrogen) containing 10% FBS, 2 mM L-glutamine and 100 IU ml$^{-1}$ penicillin/streptomycin. Mycoplasma infection was tested regularly with a PCR-based kit (MP0040, Sigma). 293T cells were used for lentivirus production. The transfection was performed by Lipofectamine 2000 according to the manufacturer's protocol (Invitrogen). For the western blot analyses, protein extracts were prepared in Triton X-100 lysis buffer[14]. Antibodies against Snail (3,895 s, Cell Signalling, mouse monoclonal, L70G2, 1:2,000), Slug (9585S, Cell Signalling, rabbit, C19G7, 1:1,000), PFKP (sc130227, Santa Cruz, rabbit polyclonal, C-23, 1:1,000), PFKM (HPA002117, Sigma, rabbit polyclonal, 1:1,000), PFKL (sc130226, Santa Cruz, rabbit polyclonal, C-22, 1:1,000), G6PD (sc373887, Santa Cruz, mouse monoclonal, G-6, 1:1,000), Flag (F-3156, Sigma, mouse monoclonal, 1:5,000), HA (12177700, Roche, mouse monoclonal, 12CA5, 1:5,000), O-GlcNAc (MMS-248R, Covance, mouse monoclonal, CTD110.6, 1:1,000) and tubulin (LF-PA0146, AbFrontier, rabbit polyclonal, 1:5,000) were obtained from the commercial vendors. The phos-tag reagents were purchased from Wako Chemicals, and gels containing phos-tag were prepared according to the manufacturer's instructions. DCF was purchased from Molecular Probe and all other chemicals were purchased from Sigma-Aldrich unless otherwise indicated.

**Plasmids and stable cell lines using lentivirus.** Tetracyclin-inducible Snail expression vector was generated with the pTRIPZ lentiviral system (Open Bio-systems) by replacing RFP with Flag-tagged Snail[13]. Flag-tagged PFKP expression vector was generated from cDNA library (Addgene 23869). Ha-tagged Wnt1 retroviral expression vector was described previously[37]. Lentivirus-mediated (pLL3.7-dsRed) Snail knockdown was described previously[48]. The Tet-pLKO-puro vector (#21915 obtained from Addgene) was used for inducible shRNA knockdown. The target sequences of shRNA were 5′-ggacgagaggagatttcaaga for shPFKP, 5′-gcaacgtagctgtcatcaacg for human shPFKP-2, 5′-gatgcacatccgaagccac for shSnail, 5′-ccactcagatgtcaagaagta for shSnail-2, 5′-gtcctgtcccagagcttattgg for shG6PD-2 and 5′-cctaaggttaagtcgccctcg for shControl-2. A pLKO-Slug-shRNA vector (#10903) was obtained from Addgene. Double strand siRNA oligo for transient knockdown of G6PD were purchased from Santa Cruz. For RNAi resistant flag-tagged PFKP expression vector, the target site of shPFKP was mutated without affecting amino acid sequences (5′-ggatgaacgccgtttccagga, mutation underlined).

**Cell death and clonogenic survival capacity assay.** For the glucose starvation experiments, $5 \times 10^3$ cells were plated into 6-well plates with normal culture medium a day before starvation. The cells were washed with PBS and cultured in glucose-free DMEM containing 10% dialysed FBS (Invitrogen). The MDA-MB-231 cells and MCF-7 cells were starved for 48 and 72 h, respectively. Cell death induced by glucose starvation was measured by trypan blue exclusion assay. Separately, clonogenic survival was performed by exposing cells to glucose starved condition followed by further observation in normal culture medium for 14 days[49]. After crystal violet (0.5% w/v) staining, colonies of more than 50 cells were counted under stereomicroscope. The results of clonogenic assay are expressed as the ratios of the number of survival colonies compared with control. For clonogenic survival against paclitaxel treatment, $5 \times 10^3$ cells were plated into 6-well plates with normal culture medium 48 h before paclitaxel treatment in absence or presence of doxycycline for inducible expression of Snail or PFKP shRNA. The cells were then cultured in paclitaxel-containing culture medium for 48 h followed by refreshment of normal culture medium for an additional 10–14 days to determine clonogenic survival. The number of colonies in 5 randomly chosen fields was determined under high power stereomicroscope. For ECM detachment assays, the cells were plated on poly-HEMA (2-hydroxyethyl methacrylate)-coated plates with complete culture medium for 24 h, and cell death then measured by trypan exclusion. The results of cell death by glucose starvation and matrix detachment are expressed as the ratio of the number of dead cells to the number of total cells counted. Cell growth was analysed by MTT (3-(4,5-dimethylthiazol-yl)-2,5-diphenyltrazolium bromide) reduction and absorbance of formazan crystals dissolved in dimethyl sulfoxide was measured at 595 nm. Cell number was counted with an EVE automatic counter (NanoEnTek).

**Quantitative PCR.** Total RNA was isolated using TRIzol reagent (Invitrogen) following the manufacturer's protocol. The SuperScript III synthesis kit (Invitrogen) was used to generate cDNA. Real-time quantitative PCR analysis for PFK-1 isoform transcripts was performed with an ABI-7300 instrument under standard conditions and SBGR mix ($n = 3$). The expression of ΔCt value from each sample was calculated by normalizing with GAPDH. Primer specificity and PCR process were verified by dissociation curve after PCR reaction. Sequence information for qPCR primers are summarized in Supplementary Information.

**Metabolic analysis and PFK kinase assay.** The intracellular levels of NADPH and total NADP (NADPH and NADP$^+$) were measured using previously reported enzymatic cycling methods[4]. In brief, cells ($5 \times 10^5$) were lysed in 200 µl of extraction buffer (20 mM nicotinamide, 20 mM NaHCO$_3$, 100 mM Na$_2$CO$_3$) and

supernatant was incubated at 60 °C for 30 min. Next, 160 µl of NADP-cycling buffer (100 mM Tris-HCl pH 8.0, 0.5 mM thiazolyl blue, 2 mM phenazine ethosulfate, 5 mM EDTA, 1.3 U of G6PD) was added to a 96-well plate containing 20 µl of the cell lysate. After incubation in the dark at 30 °C, 20 µl of 10 mM glucose-6-phophate was added to the mixture, and the change in absorbance at 570 nm was measured every 30 s for 4 min at 30 °C. The concentration of NADP$^+$ was calculated by subtracting (NADPH) from (total NADP). The intracellular level of H$_2$O$_2$ was measured with DCF (Molecular Probes). Briefly, cells ($1 \times 10^5$) were treated with DCF for 30 min. The cells were then washed with PBS and collected as single-cell suspensions. Fluorescence was detected by flow cytometry. Lactate level PFK-1 kinase activities were measured using colorimetric assay kit according to the manufacturer's protocol (Biovision). The results are expressed as the ratios of experiment to control from three independent experiments. The oxygen consumption rates (OCRs) were measured using the Seahorse XF-24 instrument (Seahorse Bioscience) under standard conditions and after the addition of 1 µM oligomycin, 0.25 µM FCCP and 0.5 µM rotenone/antimycin A. Real-time measurements of the OCR in pmol per minute per µg per protein in adjusted base medium (L-glutamine 2.05 mM, D-glucose 2 gl$^{-1}$) were plotted over time before the addition of rotenone/antimycin A, and after addition of rotenone/antimycin A to specifically measure non-mitochondrial respiration. The difference in OCR between initial non-drug treated and after addition of rotenone/antimycin A reflects the basal oxygen consumption by mitochondria. The OCR measurements were normalized with protein abundance of plated cells through BCA assay at a wavelength of 562 nm.

**$^1$H NMR and mass spectrometric analysis of metabolites.** After the culture medium was removed from the culture dish, MDA-MB-231 cells ($2 \times 10^7$) were quickly washed with ice-cold PBS. Cells were then detached from the culture dish by treatment of trypsin and quenched with HPLC grade 100% cold methanol ($-20$ °C) followed by freeze-drying. Intracellular metabolites were extracted with a solvent composed of methanol, chloroform and water (4/4/2.85, v/v/v)[50]. The aqueous phase containing water soluble metabolites was collected and freeze-dried. For NMR analysis, cell extract was resuspended in 300 µl of 100 mM sodium phosphate buffered deuterium oxide (pH 7.0) containing 0.1 mM 3-(trimethylsilyl) propionic-2,2,3,3-d$^4$ acid (TSP-d$_4$) and transferred into Shigemi NMR tubes matched with D$_2$O (Shigemi). $^1$H NMR spectra were acquired on a VNMRS 600-MHz NMR using a triple-resonance HCN salt-tolerant cold probe (Agilent Technologies) and an Ascend 800-MHz NMR using a triple-resonance 5 mm cryogenic probe (Bruker BioSpin AG). All NMR spectra were phased and baseline corrected using Chenomx NMR suite version 6.1 (Chenomx). The spectra were then normalized to the total spectral area and aligned using the correlation optimized warping (COW) method. Multivariate analysis was performed using SIMCA-P version 12.0 (Umetrics). Metabolite chemical shifts were assigned using Chenomx NMR suite version 6.1 (Chenomx Inc.) and literature[51–53], and their identities were confirmed based on total correlation spectroscopy (2D $^1$H–$^1$H TOCSY) and spiking experiments. Quantification was achieved using the 600 MHz library from Chenomx NMR Suite version 6.1 which uses the concentration of a known reference signal (TSP-d$_4$) to determine the concentration of individual compounds. Mann–Whitney tests were performed to determine significant differences among group means for non-parametric statistical hypothesis test using GraphPad PRISM version 5.0 (GraphPad Software). The differences were tested on a 95% probability level ($P < 0.05$). For the [U-$^{13}$C$_6$]-glucose tracer experiment, MDA-MB-231 cells were cultured in the RPMI1640 medium containing 0.2% $^{13}$C$_6$-labelled glucose for 24 h. Polar metabolites were extracted from cells with a solvent composed of methanol, distilled water and chloroform. The extracts were subjected 1D $^1$H{$^{13}$C} HSQC NMR analysis[54]. For mass spectrometric analysis of PPP metabolites R-5-P (ribose-5-phosphate/ribulose-5-phosphate), the cell pellets ($1 \times 10^7$ cells, MDA-MB-231) were resuspended with 200 µl of M.A.D. buffer (methanol:acetonitrile:DW = 5:3:2, v/v/v) and briefly vortexed. The mixtures were then centrifuged at 28,000 g for 20 min at 4 °C to remove the pellets. Approximately 180 µl of supernatant was centrifuged once again and 170 µl of supernatant was then evaporated to dryness by SpeedVac. The dried samples were solubilized in 20 µl of M.A.D. buffer and 2 µl injected into HPLC-ESI-MS/MS. HPLC-MS/MS analysis was performed with an Agilent 1100 Series HPLC system coupled to a Hybrid Q-Trap API 2000 mass spectrometer (AB SCIEX) with electrospray ionization (ESI) in the negative mode. R-5-P was detected with MRM (multiple reaction monitoring)[55].

**PFKP promoter assay and chromatin immunoprecipitation.** To analyse the PFKP promoter activity, the promoter region ($-539 \sim +36$ from the transcription starting site) was amplified from genomic DNA of MCF-7 cells and subcloned into pGL3 basic vector (Promega). E-box sequence 5′-CACCTG was mutated to 5′-AACCTA. To examine the PFKP promoter activity, the breast cancer cells were transfected with 100 ng of reporter vectors and 5 ng of pSV-Renilla expression vector. Luciferase and Renilla activities were measured using the dual-luciferase reporter system kit (Promega), and the luciferase activity was normalized with Renilla activity. The results are expressed as the averages of the ratios of the reporter activities from triplicate experiments. For ChIP analysis, cells were cross-linked with 1% (v/v) formaldehyde at room temperature, suspended in a ChIP lysis buffer containing 10 mM HEPES [pH 7.9], 0.5% NP-40, 1.5 mM MgCl$_2$, 10 mM KCl, 0.5 mM DTT and protease inhibitor cocktail on ice[56]. Following

centrifugation, the pellets were resuspended in buffer containing 20 mM HEPES [pH 7.9], 25% glycerol, 0.5% NP-40, 0.42 M NaCl, 1.5 mM MgCl$_2$, 0.2 mM EDTA and protease inhibitors followed by extraction of nuclear proteins. Nuclear pellets were resuspended in IP buffer (1% Triton X-100, 2 mM EDTA, 20 mM Tris-HCl [pH 8.0], 150 mM NaCl and protease inhibitors) and sonicated to break chromatin into fragments with an average length of 0.5–1 kb. The supernatants were incubated overnight with anti-Snail antibody or corresponding normal IgG. After incubation with protein G-agarose beads (Invitrogen) for 2 h at 4 °C, the beads were extensively washed with lysis buffer. Bound DNA-protein complexes were eluted following an overnight incubation at 65 °C in TE buffer. To amplify the PFKP and E-cadherin promoter fragments, the following primers were used for RT-PCR analysis: E-cadherin, forward 5′-ataacccacctagaccctagcaac, reverse 5′-ctcacaggtgctttgcagttc; PFKP, forward 5′-ctagagcccccaaccagagt, reverse 5′-gtgtgggcaggagcatctac. Separately, the cells were transfected with PFKP reporter construct (1 μg) of wild type (wt) or of E-boxes mutant (3xMut) followed by ChIP analysis to examine specific binding of Snail onto E-box. Following pull-down with Snail antibody as described above, the fold enrichment of transfected PFKP promoter fragment of wt or 3xMut reporter was detected using quantitative PCR with primers for exogenous pGL3 reporter (RVprimer3, 5′-ctagcaaaataggctgtccc) and PFKP promoter (5′-actctggttgggggctctag). The results were normalized relative to input activities and presented as mean ± s.d. from three independent experiments.

**Xenograft and lung metastasis assay.** All animal experiments were performed in accordance with the Institutional Animal Care and Use Committee of the Yonsei University and approved by the Animal Care Committee of the Yonsei University College of Dentistry and National Cancer Center Research Institute. Female athymic nude mice (6-weeks old) were used for orthotopic xenograft assays into the mammary fat pads and lung metastasis assay. For loss of function study with Snail knockdown or PFKP overexpression, MDA-MB-231 cells of control or experimental cells were harvested with trypsin treatment and injected orthotopically into the mammary fat pads ($1 \times 10^6$ per 0.1 ml of PBS) without extracellular matrix. For gain of function with transient PFKP knockdown, the MDA-MB-231 cells and metastatic clone expressing Tet-inducible PFKP shRNA were treated with control or doxycycline (5 μg ml$^{-1}$) for 48 h before injection, and further intra-peritoneal administration of doxycycline three times (at day 0, 2, 4) after tumour inoculation at a dose of 10 mg kg$^{-1}$. A xenograft assay for transient knockdown of PFKP was performed with a suboptimal number ($1 \times 10^5$ per 0.1 ml of PBS) of MDA-MB-231 cells. The tumour initiating capacity was measured twice a week using a digital caliper, and the tumour volume was calculated with the equation $V$ (in mm$^3$) $= (a \times b^2)/2$, where $a$ is the longest and $b$ is the shortest diameter. Cell proliferation and death from paraffin-embedded tumour tissue were measured with immunohistochemical staining using anti-Ki-67 (MA5-14520, SP6, rabbit poly-clonal, 1:100, Invitrogen) and active Caspase-3 (559565, C92-605, rabbit poly-clonal, 1:100, BD Bioscience) antibodies, respectively. After staining with Ki-67, digital images were taken under the light microscope (× 200). Stained cells in those images were adjusted for colour threshold and Ki-67 positive proliferating cells were counted with the ImageJ program. For lung metastasis assays, MDA-MB-231-luc-D3H2LN cells were injected into lateral tail vein, detailed experimental conditions described in each figure legend. Lung colonization was monitored and quantified using pathological examination at day 28. Mice were killed and perfused with 4% paraformaldehyde and lungs were extracted for paraffin-embedding. Paraffin-embedding sections were stained with routine haematoxylin and eosin and the number of lung metastatic nodules was counted under microscopic examination.

**Statistical analysis and reproducibility.** All statistical analysis of cell death, clonogenic, NADPH and lactate was performed with two-tailed Student's $t$-tests; data are expressed as means and s.d. The **denote $P < 0.01$, *$P < 0.05$. Statistical significance of animal experiments and metabolomic analysis was determined using the Mann–Whitney test. No statistical method was used to predetermine sample size.

**Data availability.** Metabolomics NMR data that support the findings of this study have been deposited in MetaboLights (http://www.ebi.ac.uk/metabolights) under the accession codes MTBLS387. All other data are available within the manuscript or from the corresponding authors upon reasonable request.

The authors declare that all the data supporting the findings of this study are available within the article and its Supplementary information files and from the corresponding author upon reasonable request.

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

## Acknowledgements

We thank Professor S.M. Jeon, H.G. Yoon, H. Ko, Y.M. Huh and the Cancer Metabolism Research Group in Korea for helpful discussion and technical assistance. We also thank E. Tunkle for preparation of the manuscript and K.Y. Kim for statistical analysis. We thank J.W. Cho for helpful discussion of *O*-glycosylation. This work was supported by grants from the National Research Foundation of Korea (NRF-2012M3A9B2052523, NRF-2014R1A2A1A05004670, NRF-2016R1E1A1A01942724) funded by the Korea government (MSIP), a grant from the National Research Foundation of Korea (NRF-2014R1A6A3A04055110) funded by the Korea government (MOE), and a grant from the National R&D Program for Cancer Control, Ministry for Health, Welfare and Family Affairs, Republic of Korea (1420310). This work was also supported by the by the National Research Council of Science & Technology (CAP-2012-2-KBSI), and the Korea Basic Science Institute (C36705). S.Y. Kim was supported by a grant from the National Cancer Center of Korea (NCC1410670).

## Author contributions

N.H.K. and Y.H.C. involved and performed most of the experiments. J.L., M.N., N.K., S.H.P. and K.S.H. performed metabolomics analysis. J.H.Y., J.S.Y. and J.K.R. prepared expression vectors and shRNA constructs. X.Z. and E.S.C. performed immunohisto-chemical works. S.Y.C., Y.L. and Y.S.Y. performed *in vivo* experiments. S.H.L. and S.Y.K. performed independent *in vivo* experiments. S.W.K. support ROS experiments and provide comments. J.I.Y., J.H.J., H.S.K. and G.S.H. conceived the study and wrote the paper. All authors read and approved the manuscript.

## Additional information

**Competing financial interests:** The authors declare no competing financial interests.

