## [Peer Review File · Nature Communications]

REVIEWERS' COMMENTS:

Reviewer #2 (Remarks to the Author):

The authors have provided the data I requested. However, I have some concerns about the other responses.

The response to the concern of reviewer 1 seems unsatisfactory. As I understand, the authors are claiming they load 40ug and 8ug protein from MCF7 and MDA MB231 cells to look for Snail expression, but 20ug of protein as a loading control? I do not think this is acceptable. Also, the fact that the cell lines are not on a list really doesn't help tell us whether the lines the authors are using are what they expect. Authentication of their lines would still be necessary.

In response to reviewer 3, I don't think revising the title of the paper overcomes the need to show that there is increased diversion of glucose into the PPP.

Reviewer #3 (Remarks to the Author):

The authors have addressed my concerns.

Reviewer #4 (Remarks to the Author):

I was asked to comment on questions raised about metabolic flux analysis in the paper by Kim et al.

The authors indicate that they have performed flux (actually isotope incorporation) measurements to show alterations in glycolysis and mass spectrometry (MS) to measure increased utilization of the PPP pathway. The NMR data do show a reduction in glycolysis as well as changes in a number of amino acids are related metabolites. While it is true that NMR is not able in this experiment to measure PPP metabolites (it can do so when sufficient sample is available), the authors provide MS data showing an increase in ribulose-5-phosphate, which they interpret as proof of a diversion of flux from glycolysis and lactate production. While I am generally of the opinion that isotope tracing experiments provide stronger proof of altered metabolite flux, the authors do observe similar changes to a previous paper (Ref 30, Yamamoto, Nat. Comm 2014), which I think is sufficient proof of the present authors' claims at this stage.

Nevertheless, there are a couple of minor improvements to the manuscript that should be made prior to publication:

- 1) It is not always clear in the figures or text exactly which cell type are being used, especially for the metabolomics and pathway measurements. The authors should either add annotation to their figures captions or in the text.
- 2) Did the authors perform normalization prior to reporting the metabolite measurements? What type? Unless the data are normalized correctly, the results could be misleading.

Point-by-point response to reviewers' comments (NCOMMS-16-10755B)

Reviewer #2 (Remarks to the Author):

The authors have provided the data I requested. However, I have some concerns about the other responses. The response to the concern of reviewer 1 seems unsatisfactory. As I understand, the authors are claiming they load 40ug and 8ug protein from MCF7 and MDA MB231 cells to look for Snail expression, but 20ug of protein as a loading control? I do not think this is acceptable. Also, the fact that the cell lines are not on a list really doesn't help tell us whether the lines the authors are using are what they expect. Authentication of their lines would still be necessary. In response to reviewer 3, I don't think revising the title of the paper overcomes the need to show that there is increased diversion of glucose into the PPP.

(Author's Response) Heeding the reviewer's point, we have re-performed immunoblot analysis for endogenous Snail using the same amount of cell lysate (20 ug) and detected control from the same blot (Fig. 1a, Fig. 3b, Fig. 6a, and Fig. 7b). We also added a detailed description of the immunoblot analysis for endogenous Snail in Figure legends. Further, we have performed authentication of cell lines using short tandem repeat (STR) analysis following the guidelines of the American Tissue Culture Collection Standards Development Organization Workgroup (Supplementary Fig. 11).

Reviewer #3 (Remarks to the Author):

The authors have addressed my concerns.

Reviewer #4 (Remarks to the Author):

I was asked to comment on questions raised about metabolic flux analysis in the paper by Kim et al.

The authors indicate that they have performed flux (actually isotope incorporation) measurements to show alterations in glycolysis and mass spectrometry (MS) to measure increased utilization of the PPP pathway. The NMR data do show a reduction in glycolysis as well as changes in a number of amino acids are related metabolites. While it is true that NMR is not able in this experiment to measure PPP metabolites (it can do so when sufficient sample is available), the authors provide MS

data showing an increase in ribulose-5-phosphate, which they interpret as proof of a diversion of flux from glycolysis and lactate production. While I am generally of the opinion that isotope tracing experiments provide stronger proof of altered metabolite flux, the authors do observe similar changes to a previous paper (Ref 30, Yamamoto, Nat. Comm 2014), which I think is sufficient proof of the present authors' claims at this stage.

Nevertheless, there are a couple of minor improvements to the manuscript that should be made prior to publication:

1) It is not always clear in the figures or text exactly which cell type are being used, especially for the metabolomics and pathway measurements. The authors should either add annotation to their figures captions or in the text.

2) Did the authors perform normalization prior to reporting the metabolite measurements? What type? Unless the data are normalized correctly, the results could be misleading.

(Author's Response) We thank the reviewer for the helpful comments. We have added the cell type for metabolomics and pathway measurements to the text and figure legends. Following the reviewer's advice, we added a description of the data normalization in the Methods section.